# Focused Ultrasound-Mediated Delivery of Anti-Programmed Cell Death-Ligand 1 Antibody to the Brain of a Porcine Model

**DOI:** 10.3390/pharmaceutics15102479

**Published:** 2023-10-17

**Authors:** Siaka Fadera, Chinwendu Chukwu, Andrew H. Stark, Yimei Yue, Lu Xu, Chih-Yen Chien, Jinyun Yuan, Hong Chen

**Affiliations:** 1Department of Biomedical Engineering, Washington University in St. Louis, St. Louis, MO 63130, USA; s.fadera@wustl.edu (S.F.); yue.y@wustl.edu (Y.Y.); jinyun.yuan@wustl.edu (J.Y.); 2Department of Neurosurgery, Washington University School of Medicine, St. Louis, MO 63110, USA

**Keywords:** focused ultrasound, immunotherapy, immune checkpoint inhibitors, blood-brain barrier, brain drug delivery

## Abstract

Immune checkpoint inhibitor (ICI) therapy has revolutionized cancer treatment by leveraging the body’s immune system to combat cancer cells. However, its effectiveness in brain cancer is hindered by the blood-brain barrier (BBB), impeding the delivery of ICIs to brain tumor cells. This study aimed to assess the safety and feasibility of using focused ultrasound combined with microbubble-mediated BBB opening (FUS-BBBO) to facilitate trans-BBB delivery of an ICI, anti-programmed cell death-ligand 1 antibody (aPD-L1) to the brain of a large animal model. In a porcine model, FUS sonication of targeted brain regions was performed after intravenous microbubble injection, which was followed by intravenous administration of aPD-L1 labeled with a near-infrared fluorescent dye. The permeability of the BBB was evaluated using contrast-enhanced MRI in vivo, while fluorescence imaging and histological analysis were conducted on ex vivo pig brains. Results showed a significant 4.8-fold increase in MRI contrast-enhancement volume in FUS-targeted regions compared to nontargeted regions. FUS sonication enhanced aPD-L1 delivery by an average of 2.1-fold, according to fluorescence imaging. In vivo MRI and ex vivo staining revealed that the procedure did not cause significant acute tissue damage. These findings demonstrate that FUS-BBBO offers a noninvasive, localized, and safe delivery approach for ICI delivery in a large animal model, showcasing its potential for clinical translation.

## 1. Introduction

It is estimated that over 300,000 individuals are diagnosed with primary brain tumors annually, with an even larger number affected by brain metastases. Among the recent groundbreaking advancements, immune checkpoint inhibitor (ICI) therapy has emerged at the forefront, offering a promising avenue to bolster the body’s defense mechanisms against tumor cells. ICIs are therapeutic agents that work by blocking proteins that inhibit the immune system, enabling the immune cells to recognize and attack cancer cells more effectively [1]. Several ICIs have been approved for the treatment of a range of cancers, including melanoma, non-small-cell lung cancer, head and neck cancer, and urothelial carcinoma [2,3,4]. The great success of ICIs in treating solid tumors has led to several ongoing clinical trials exploring their efficacy in primary or recurrent brain tumor patients. However, the therapeutic efficacy reported to date for brain tumors has been poor [5,6,7]. Although the current evidence supporting the reason for these poor outcomes remains ambivalent, an increasing number of reports suggest that more efficient targeting of the immune cells in the tumor microenvironment using ICIs could lead to better therapeutic outcomes [8,9]. However, the blood-brain barrier (BBB) restricts the entry of ICIs into the brain tumor microenvironment [10].

Different approaches have been proposed for overcoming the BBB for ICI delivery to the brain. Guo et al. conjugated an ICI, anti-programmed death ligand 1 antibody (aPD-L1), with a targeting moiety p-hydroxybenzoic acid to cross the BBB based on the dopamine-receptor-mediated transcytosis [11]. They reported that this conjugation achieved 2-fold higher aPD-L1 delivery to the brain of GL261 glioma-bearing mice than unmodified aPD-L1. The enhanced delivery contributed to a significantly longer median survival time. Wang et al. fabricated aPD-L-conjugated copolymer nanoparticles that can cross the BBB through receptor-mediated transportation [12]. This nanomedicine achieved efficient aPD-L1 delivery and triggered effective immune responses that significantly prolonged the survival of mice implanted with glioma. Receptor-mediated trans-BBB delivery is limited by the need to conjugate ICI with BBB-permeable agents, and it cannot achieve targeted delivery, specifically to brain tumors. Focused ultrasound combined with microbubble-mediated BBB opening (FUS-BBBO) is an emerging technique that can overcome the limitations of existing approaches to achieve spatially targeted drug delivery without needing customized drugs.

FUS-BBBO is an emerging technique that utilizes transcranial low-intensity FUS combined with intravenously injected microbubbles for the noninvasive, spatially targeted, and reversible opening of the BBB [13,14]. FUS can penetrate the skull noninvasively and focus on virtually any brain region with millimeter-scale accuracy. Microbubbles, gas-filled micron-sized bubbles coated by shells, have been used in the clinic as blood-pool contrast agents for ultrasound imaging. Combined with FUS, they amplify and localize FUS-mediated mechanical effects on the cerebral vasculature to induce BBB opening. Extensive preclinical research has been conducted to develop and evaluate this technique [15,16,17]. Recent clinical studies have demonstrated the feasibility and safety of FUS-BBBO in patients with various brain diseases, including amyotrophic lateral sclerosis, Alzheimer’s disease, Parkinson’s disease, and glioma [18,19,20,21]. There are three clinical trials underway using the FUS-BBBO technique for the delivery of ICIs in patients with recurrent glioblastoma (clinicaltrial.gov ID: NCT05879120) and brain metastases (NCT05317858 and NCT05879120) [22]. However, as far as we known, only one preclinical study has been reported on the FUS-BBBO delivery of ICIs. Lee et al. reported an improved delivery of an ICI to the glioblastoma tumor using FUS-BBBO in mice, which extended mouse survival [23]. However, no large animal studies have been reported that evaluate the feasibility and safety of ICI delivery via FUS-BBBO. Large animal studies are critically needed to obtain the data required to support the clinical application of this promising technique.

Here, we report the first large animal study of FUS-BBBO delivery of aPD-L1 via FUS-BBBO. The FUS-BBBO procedure was performed in pigs, which was followed by intravenous injection of near-infrared fluorescent dye-labeled aPD-L1. The BBB permeability change was evaluated using contrast-enhanced MRI. aPD-L1 delivery outcome was evaluated via fluorescence imaging. The safety of the procedure was assessed via in vivo MRI imaging combined with ex vivo histological staining of brain slices.

## 2. Materials and Methods

### 2.1. Synthesis and Characterization of Near-Infrared Fluorescence Dye-Labeled aPD-L1

The synthesis and characterization of near-infrared fluorescent dye-labeled aPD-L1 was performed in accordance with our previously reported procedure [24] with modifications. We obtained the infrared dye 800CW (IRDye 800CW) with an NHS ester from LI-COR Biosciences (Lincoln, NE, USA). The IRDye 800-aPD-L1 conjugation was performed according to the manufacturer’s instructions for high-molecular-weight protein labeling. Briefly, the pH of aPD-L1 was adjusted to above seven using 1 M potassium phosphate (pH 9). The aPD-L1 and dye were mixed at 2:1 dye:aPD-L1 mol/mol, while the mixture was protected from light via incubation at room temperature for a minimum period of 2 h. After incubation, a 1xPBS pre-equilibrated desalting column (7000 MWCO, Thermo Scientific, Waltham, MA, USA) was used to purify the labeled aPD-L1. The final product of 800CW-aPD-L1 was kept at 4 °C for further applications. A NanoDrop (Thermo Fisher Scientific, Waltham, MA, USA) was used to determine the amount of protein conjugated to the dye at absorbances of 280 nm (A280) and 780 nm (A780) according to the manufacturer’s instructions (LI-COR Bioscience).

### 2.2. Animals

The Institutional Animal Care and Use Committee of Washington University in St. Louis reviewed and approved all animal procedures in accordance with the National Institute of Health Guidelines for animal research. We performed all the procedures on wild-type pigs (breed: Yorkshire white, age: 4 weeks, sex: male, weight: 15 lbs., Oak Hill Genetics, Ewing, IL, USA). Intramuscular injections of ketamine (2 mg/kg), xylazine (2 mg/kg), and telazol (4 mg/kg) were used to sedate the pigs and isoflurane was used to keep the pigs under general anesthesia while intubated. A hair removal cream (Nair, Church & Dwight Co., Princeton, NJ, USA) was used to remove the hair on the pigs’ heads in preparation for FUS sonication. An intravenous (IV) catheter was placed in the lateral saphenous vein in order to inject microbubbles and gadolinium.

### 2.3. FUS Setup and FUS-BBBO Procedure

The FUS device (Image Guided Therapy, Bordeaux, France) included a 15-element annular-ring FUS transducer (Imasonic, Voray sur l’Ognon, France; center frequency of 650 kHz, aperture of 65 mm, and focal length of 65 mm), electrical driving system, a 3D motor system for transducer positioning, and a computer control system. The acoustic pressure fields generated by the FUS transducer were calibrated using a hydrophone (HGL-0200; Onda Inc., Sunnyvale, CA, USA) in a degassed water tank. The axial and lateral full width at half maximum (FWHM) dimensions of the FUS transducer were 22.0 mm and 3.3 mm, respectively. The peak negative pressures of the FUS transducer under different input voltage levels were measured at the transducer’s focus in the water tank.

During the FUS-BBBO procedure, pigs were positioned on a custom-made head frame with head supports to stabilize their head and ensure it remained flat (Figure 1), as reported in our previous publication [25]. Degassed ultrasound gel was applied on the pig’s head, followed by the positioning of a water chamber filled with degassed water for acoustic coupling. The transducer was placed horizontally above the head to achieve an approximately 90-degree incident angle, thereby optimizing transmission efficiency through the skull. For each pig subject, three cortical regions in one brain hemisphere were sonicated, and the contralateral hemisphere was used as the nonsonicated control. The three sonicated regions were separated by 10 mm between the adjacent targets. A 10 min delay between two sonications was implemented to avoid interference between repeated treatments. The intravenous injection of microbubbles (Definity^®^, Lantheus Medical Imaging, N Billerica, MA, USA), following the clinically recommended dose of 10 µL/kg, was performed immediately prior to each FUS sonication. These microbubbles comprised octafluoropropane encapsulated in an outer lipid shell with a mean diameter within the range of 1.1 to 3.3 µm, according to the manufacturer-provided information. FUS sonication parameters were the same as reported in our previous study [26] (frequency: 0.65 MHz, free-field pressure measured in the water tank: 3 MPa, pulse repetition frequency: 1 Hz, pulse length: 10 ms, and sonication duration: 3 min). The 800CW-aPD-L1 (1.152 mg/kg) was intravenously injected immediately post-FUS sonication.

### 2.4. In Vivo MRI for BBB Permeability Evaluation and Safety Assessment

After FUS sonication, each pig was transported to a 3T Siemens Allegra MRI scanner (Siemens Medical Solutions, Malvern, PA, USA). The BBB permeability change was evaluated using a T1-weighted gradient echo sequence (TR/TE: 2300/3.33 ms; slice thickness: 0.9 mm; in-plane resolution: 0.94 × 0.94 mm^2^; matrix size: 192 × 192; flip angle: 27°). The scans were acquired pre- and post-intravenous injection of a chelated gadolinium-based MRI contrast agent (Multihance, Bracco Diagnostics Inc., Monroe Township, NJ, USA) at an injection dose of 0.2 mL/kg. We utilized T2-weighted SPACE (TR/TE: 3200/408 ms; slice thickness: 0.9 mm; in-plane resolution: 0.9 × 0.9 mm^2^; matrix size: 256 × 256; flip angle: 120°) and T2-weighted fluid-attenuated inversion recovery (FLAIR) imaging (TR/TE: 4800/442 ms; slice thickness: 0.9 mm; in-plane resolution: 0.9 × 0.9 mm^2^; matrix size: 256 × 256; flip angle: 120°) to detect edema and hemorrhage, respectively. The BBB permeability change was evaluated by calculating the MRI contrast-enhancement volume using a custom MATLAB script, as reported in our previous publication [26]. Briefly, regions of interest (ROIs) were defined as the FUS-treated region (FUS+) and contralateral region (FUS−). Voxels in the FUS+ ROI with intensities greater than 3× standard deviations above the mean intensity of the FUS− ROI were identified for each MRI slice. The contrast-enhancement volume was calculated by the sum of the identified voxels across whole brain slices.

### 2.5. Ex Vivo Fluorescence Imaging for Evaluating aPD-L1 Delivery Outcome

After MRI scans, pigs were sacrificed and their brains were harvested and fixed in 10% formalin for at least 48 h prior to cutting into 2 mm slices, obtained using a 3D-printed brain slicer matrix. The brains were cut horizontally, preserving the FUS-sonicated regions and nonsonicated contralateral regions on the same level. The fixed brain slices were imaged using the LI-COR Pearl imaging system with the 800 nm acquisition channel for 800CW-aPD-L1 imaging. All images were acquired under the same imaging parameters, including the exposure time. The fluorescence intensity of brain slices was quantified using the LI-COR Image Lite Software (version 5.2).

### 2.6. Ex Vivo Histological Safety Analysis

After fluorescence imaging of the 2 mm-thick brain slices, those slices with the highest fluorescence intensity were frozen and sectioned into 10 µm slices for hematoxylin and eosin (H&E) staining. The stained slices were imaged using a microscope (BZ-9000; Keyence, Osaka, Japan) with 2× and 20× objectives.

### 2.7. Statistical Analysis

Statistical analysis was performed using GraphPad Prism (Version 9.5.1, Boston, MA, USA). Representative data were presented as mean values ± standard deviation (SD). The statistical significance difference between the experimental groups was analyzed using the unpaired two-tailed Student’s *t*-test.

## 3. Results

### 3.1. MRI Confirmation of FUS-BBBO

Successful BBB opening in the pig brain was confirmed using T1-weighted MRI images. A significant enhancement in gadolinium extravasation was observed in all the FUS-targeted regions, as shown in Figure 2. On average, a 4.8-fold increase in the contrast-enhancement volume in the FUS-targeted regions was observed compared to that in the contralateral nontargeted regions.

### 3.2. FUS-Mediated Delivery of Fluorescence-Labeled aPD-L1

Figure 3A shows pictures of horizontal brain slices and corresponding fluorescence images. A higher localized accumulation of fluorescence-labeled aPD-L1 was observed at the FUS-targeted region than at the nontargeted region. Quantification of the fluorescence intensity, as summarized in Figure 3B, revealed a 2.1-fold average increase in fluorescence intensity in the FUS-targeted regions (FUS+) compared to the nontargeted contralateral brain regions (FUS−).

### 3.3. MRI Evaluation of Safety

T2-weighted SPACE and FLAIR were used to detect signal-intensity changes after FUS sonication. Although the image quality of the FLAIR images was relatively low, no clear sign of either hyperintensity or hypointensity was observed in the targeted brain regions after FUS sonication (Figure 4).

### 3.4. Histological Assessment of Safety

The gross pathology examination of the post-craniotomy brain surface in the three pigs revealed no obvious tissue damage induced by the FUS procedure (Figure 5A). H&E staining found microhemorrhaging in six of the nine targeted brain regions (Figure 5B), with an average dimension of 201 µm.

## 4. Discussion

This was the first study of FUS-BBBO delivery of ICIs to the brain in a large animal model. Our findings indicate that FUS-BBBO significantly enhanced the delivery of aPD-L1 at the FUS-targeted regions without causing significant acute damage to the brain. These results underscore the promising clinical translation potential of FUS-BBBO for enhancing ICI therapy for brain cancer.

FUS-BBBO has been utilized for delivering various therapeutic agents to the brain; however, its effectiveness in delivering ICIs has not been thoroughly investigated. In a previous study by Lee et al., anti-PD1 was administered using closed-loop controlled FUS sonication in a mouse model of glioblastoma [23]. Immunohistochemistry staining of ex vivo mouse brain slices following the FUS-BBBO procedure revealed a 2.5-fold increase in anti-PD1 intensity compared to that in the nonsonicated control group. In our study, performed on a pig model, the intensity of fluorescent-labeled aPD-L1 was found to be 2.1-fold higher than that in the control (Figure 2B). It is important to note that the mouse and pig studies employed different animal models and treatment parameters, making direct comparisons of the delivery outcomes difficult. Nevertheless, these findings suggest that FUS-BBBO achieved more than 2-fold enhancement in ICI delivery in mice and pigs.

We found that utilizing aPD-L1 labeled with a near-infrared fluorescent dye is a valuable method for visualizing the spatial distribution of aPD-L1. This approach proved particularly advantageous in the pig study due to the technical challenges associated with sectioning and immunohistochemistry staining of the large pig brain. The fluorescence images obtained from pig brain slices demonstrated the successful delivery of aPD-L1 in all pigs; however, variations in the delivery outcomes were observed among the three different pigs and even among the three target brain regions within the same pig (Figure 2A). These findings underscore the importance of developing a feedback-control algorithm capable of regulating the FUS-sonication parameters during the procedure, leveraging real-time measurements of the acoustic emissions from microbubbles.

In vivo MRI and ex vivo histological analysis revealed that the procedure did not cause significant acute tissue damage. Following craniotomy, petechial hemorrhages were observed on the brain surface (Figure 5A), likely resulting from vessel ruptures in the meninges layer during the craniotomy procedure. In six of the nine targets, petechial hemorrhages were also observed in the FUS-sonicated brain regions (Figure 5B). In a previous study by Huang et al., FUS sonication was considered unsafe if accumulations of extravagated red blood cells were larger than 500 µm in diameter [27]. In all the FUS-targeted brain regions in our study, the dimension of red blood cell extravasation was found to be smaller than 500 µm. Furthermore, the microscopic leakage of red blood cells was not detectable in the T2W and FLAIR images (Figure 4). It is important to note that these safety assessments were conducted within 4 h after FUS sonication. Notably, microhemorrhaging and edemas were observed post-FUS sonication in several reported clinical studies of FUS-BBBO, but the problem typically resolved itself within a week or month [18,19,28].

This study demonstrated the feasibility and safety of FUS-BBBO in a large animal model, which supports the clinical translation of this technique for the delivery of ICIs. Although promising, this feasibility study has several limitations. First, this feasibility study used healthy pigs as the animal model. Future studies could further evaluate the efficiency and efficacy of drug delivery using a pig model of brain tumors [25]. Second, this study performed single-point sonication in each targeted brain location. Future studies need to implement large-volume sonication for efficient delivery of ICIs to the entire brain tumor.

In summary, FUS-BBBO is a promising technology that can potentially improve the outcomes of immune checkpoint inhibitor therapy for brain tumors by enabling the noninvasive, spatially targeted delivery of the inhibitors.

## Figures and Tables

**Figure 1 pharmaceutics-15-02479-f001:**
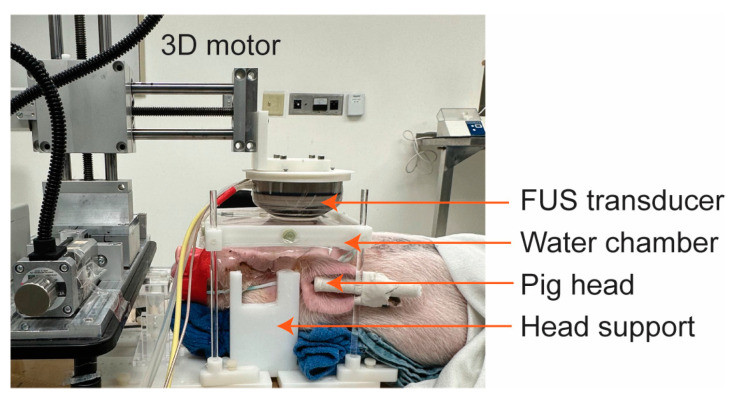
Picture of the FUS setup for pigs. The head was stabilized by head supports. The FUS transducer was positioned next to a 3D motor to target the desired brain regions. The transducer was coupled to the pig’s head using a water chamber.

**Figure 2 pharmaceutics-15-02479-f002:**
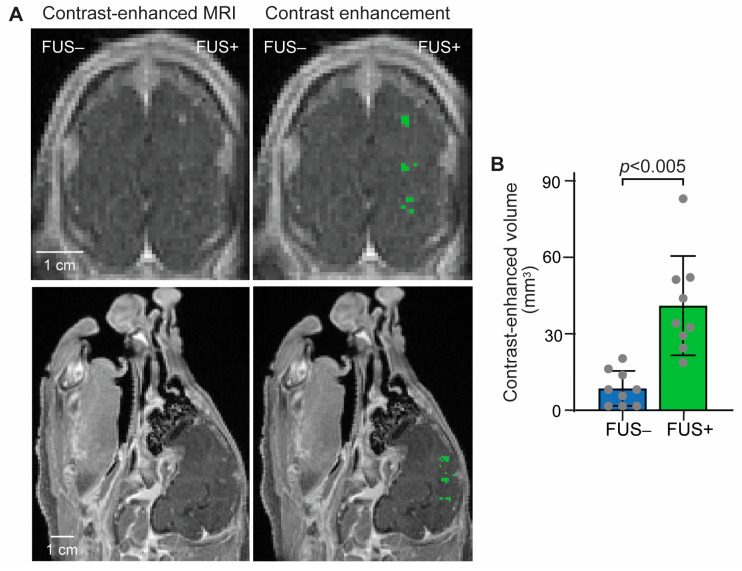
Contrast-enhanced MRI confirmed FUS-BBBO in the pig brain. (**A**) **Left**: Representative contrast-enhanced T1-weighted MRI image in the coronal and sagittal planes; **Right**: Pixels with higher contrast enhancement than the contralateral side were identified and highlighted in green. (**B**) The contrast-enhanced volume was significantly increased. Each dot represents the measurement at one targeted brain region. Error bars indicate standard deviation.

**Figure 3 pharmaceutics-15-02479-f003:**
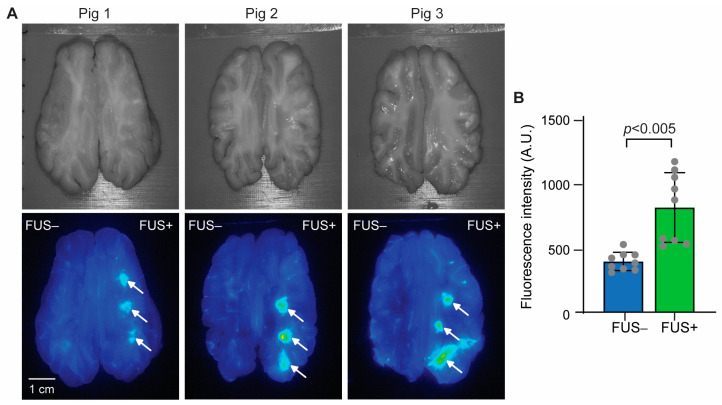
FUS-BBBO delivery of aPD-L1 to the pig brain. (**A**) **Top**: Pictures of the pig brain slices. **Bottom**: Corresponding fluorescence images of the brain slices. FUS was targeted on the right side of the brain, as indicated by arrows (FUS+), while the contralateral side was nonsonicated (FUS−). (**B**) Quantification of the fluorescence intensity for each targeted brain region.

**Figure 4 pharmaceutics-15-02479-f004:**
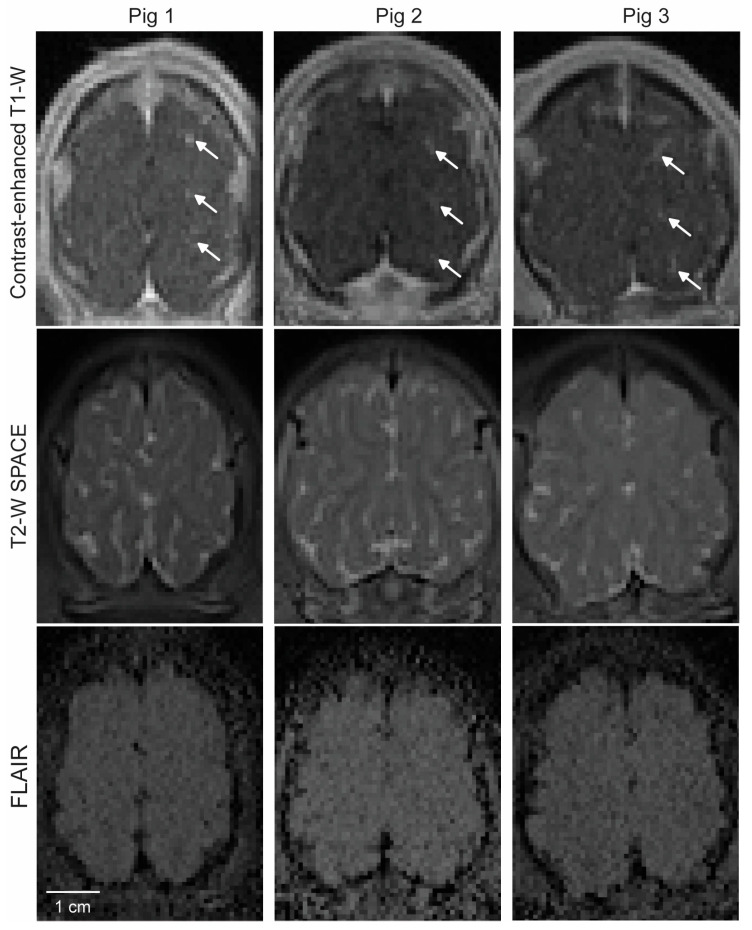
MRI safety assessment. **Top**: Contrast-enhanced MRI showing the location of BBB opening in each pig, as indicated by arrows. **Middle**: Corresponding T_2_-weighted SPACE MRI captured at the same imaging plane as in the top image. **Bottom**: Corresponding FLAIR images.

**Figure 5 pharmaceutics-15-02479-f005:**
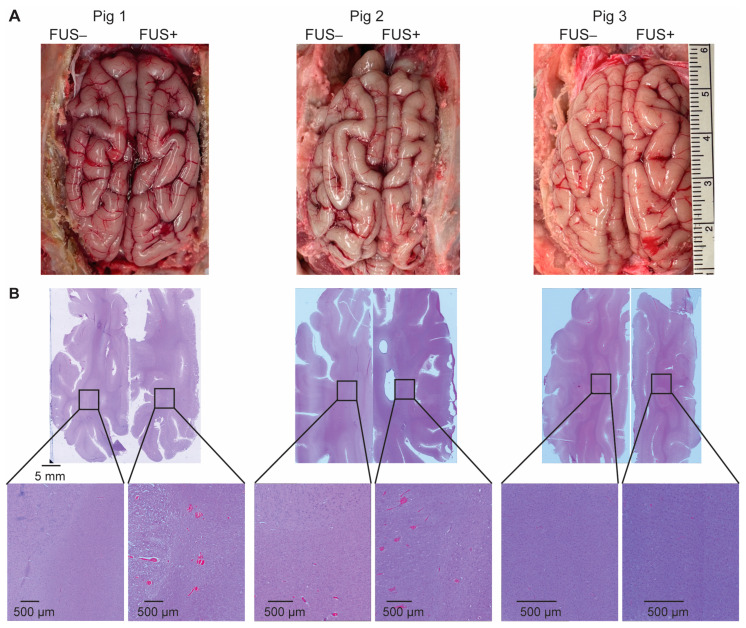
Ex vivo safety assessment. (**A**) The gross pathology examination of the post-craniotomy brain surface in the three pigs revealed no obvious tissue damage induced by the FUS procedure. (**B**) Hematoxylin and eosin (H&E) staining of the sonicated- and nonsonicated-brain tumor tissue identified microhemorrhages in six of the nine target regions.

## Data Availability

The data presented in this study are available on request from the corresponding author.

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
