# Peer review of "Focused Ultrasound-Mediated Delivery of Anti-Programmed Cell Death-Ligand 1 Antibody to the Brain of a Porcine Model"

_pharmaceutics, 2023, doi:10.3390/pharmaceutics15102479_

Round 1

Reviewer 1 Report

In this manuscript, the authors assessed the safety and feasibility of using focused ultrasound combined with microbubbles-mediated BBB opening (FUS-BBBO) to facilitate trans-BBB delivery of an ICI, anti-programmed cell death-ligand 1 antibody (aPD-L1), to the brain of a large animal model. However, there are still the major concerns that need to be addressed.

1. The authors claimed to have used intravenous microbubble injections for FUS-BBBO in porcine brain, yet the manufacture and characterization of the microbubbles is not mentioned. Moreover, the purpose of the microbubbles is not explained.

2. In my experience, the power of ultrasound has a significant impact on the safety of the therapy, and the authors should have given the power parameters of ultrasound.

3. The authors should have assessed the efficacy of aPD-L1, because FUS has mechanical and cavitation effects, which may be potentially damaging to aPD-L1.

4. The pictures show that the contrast-enhanced volume changed significantly before and after FUS administration, but the formula for its calculation should be given.

5. Some related references should be cited, such as Progress in Materials Science, 2023, DOI: 10.1016/j.pmatsci.2023.101170; Journal of Pharmaceutical Analysis, 2023, 13(3), 239-254; Journal of Advanced Research, 2023, 43, 87-96.

None

Author Response

Responses to Reviewer #1

Comment 1: The authors claimed to have used intravenous microbubble injections for FUS-BBBO in porcine brain, yet the manufacture and characterization of the microbubbles is not mentioned. Moreover, the purpose of the microbubbles is not explained.

Response:

This study used commercially available microbubbles that have been used in the clinic as ultrasound contrast agents. In the revised manuscript, we added the information of the microbubble manufacture and characterization and the purpose of using the microbubbles.

Manuscript Revision:

“Microbubbles, gas-filled micron-sized bubbles coated by shells, have been used in the clinic as blood-pool contrast agents for ultrasound imaging. When combined with FUS, they amplify and localize FUS-mediated mechanical effects on the cerebral vasculature to induce BBB opening.” [Lines 64-67]

“Intravenous injection of microbubbles (Definity®, Lantheus Medical Imaging, N Billerica, MA, USA) following a clinically recommended dose of 10 µL/kg was performed immediately prior to each FUS sonication. These microbubbles were octafluoropropane encapsulated in an outer lipid shell with a mean diameter within the range of 1.1 to 3.3 µm.” [Lines 134-138]

Comment 2: In my experience, the power of ultrasound has a significant impact on the safety of the therapy, and the authors should have given the power parameters of ultrasound.

Response:

For thermal-based therapeutic applications of ultrasound, power is an important parameter; however, for the focused ultrasound-mediated blood-brain barrier technique, ultrasound-induced microbubble cavitation is the main mechanism. Ultrasound pressure is a more relevant parameter than power for this technique.

Comment 3: The authors should have assessed the efficacy of aPD-L1, because FUS has mechanical and cavitation effects, which may be potentially damaging to aPD-L1.

Response: 

In this study, aPD-L1 was injected after FUS sonication. There was no direct interaction between FUS and aPD-L1.

Comment 4: The pictures show that the contrast-enhanced volume changed significantly before and after FUS administration, but the formula for its calculation should be given.

Response:

We have revised the manuscript to add the details for the quantification methods.

Manuscript Revision: 

“The BBB permeability change was evaluated by calculating the MRI contrast-enhancement volume using a custom MATLAB script as reported in our previous publication. Briefly, regions of interest (ROIs) were defined at the FUS treated (FUS+) region and contralateral region (FUS-). Voxels in the FUS+ ROI had intensities greater than 3× standard deviations above the mean intensity of the FUS- ROI were identified for each brain MRI slice. The contrast enhancement volume was calculated by the sum of the identified voxels across whole brain slices.” [Lines 159-165]

Comment 5: Some related references should be cited, such as Progress in Materials Science, 2023, DOI: 10.1016/j.pmatsci.2023.101170; Journal of Pharmaceutical Analysis, 2023, 13(3), 239-254; Journal of Advanced Research, 2023, 43, 87-96.

Response:

We have located the three mentioned references and provided their detailed citations below. We chose not to include them in our citation list as they are not directly pertinent to our research. While the reviewers can choose to reject our manuscript, it is not appropriate to leverage their position to request the addition of unrelated citations.

Li, Xin, Yue Gao, Helin Li, Jean-Pierre Majoral, Xiangyang Shi, and Andrij Pich. "Smart and bioinspired systems for overcoming biological barriers and enhancing disease theranostics." Progress in Materials Science (2023): 101170.

Yi, Lu, Qiulan Luo, Xiaobin Jia, James P. Tam, Huan Yang, Yuping Shen, and Li, Li. Multidisciplinary strategies to enhance therapeutic effects of flavonoids from Epimedii Folium: Integration of herbal medicine, enzyme engineering, and nanotechnology, Journal of Pharmaceutical Analysis, 2023, 13(3), 239-254

Li, Xin, Laura Hetjens, Nadja Wolter, Helin Li, Xiangyang Shi, and Andrij Pich. "Charge-reversible and biodegradable chitosan-based microgels for lysozyme-triggered release of vancomycin." Journal of Advanced Research 43 (2023): 87-96.

Reviewer 2 Report

I congratulate the authors on a well conceived and executed research project demonstrated the feasibility of FUS-BBB opening to deliver ICI in pigs>

Clearly this is a preliminary report that I would like to see augmented by large animal survival studies to assess toxicity.

Author Response

Responses to Reviewer #2

Comment: Clearly this is a preliminary report that I would like to see augmented by large animal survival studies to assess toxicity.

Response:

We thank the reviewer for the comment. The focus of this study was to demonstrate the feasibility of FUS-BBB opening in a large animal model. Our future work will perform survival studies to evaluate the toxicity, but it is beyond the scope of the current study.

Reviewer 3 Report

This is a well-designed and well-written manuscript exploring the feasibility and safety of BBB disruption in large animals. It paves the way to further use the treatment strategy in clinical trials. The findings are comparable to our published study on sonodynamic therapy for GBM in large animals. However, we know the treatment trajectory of focused ultrasound. I would like the authors to show us the planes of MRI, i.e., sagittal plane or coronal planes to better appreciate the treatment envelops. 

Author Response

Responses to Reviewer #3

Comment: The findings are comparable to our published study on sonodynamic therapy for GBM in large animals. However, we know the treatment trajectory of focused ultrasound. I would like the authors show us the planes of MRI, i.e., sagittal plane or coronal plnes to better appreciate the treatment envelops.

Response:

We appreciate the reviewer for the suggestion. In our revised manuscript, we added the sagittal plane in Fig. 2.

Reviewer 4 Report

Fadera and colleagues present an interesting work, titled “Focused ultrasound-mediated delivery of anti-programmed cell death-ligand 1 antibody to the brain of a porcine model” showing the potential of FUS-BBBO to enhance brain drug delivery of anti-programmed cell death-ligand 1 antibody (aPD-L1). Also, the authors used for the first time a porcine model to investigate the therapeutical potential of FUS-BBBO while most reports have used rats and mice.

Overall, the study was well-conducted and present value to our understanding of the therapeutical potential of the focused ultrasound-mediated delivery through the BBB. However, some points need to be addressed to enhance the quality and clarity of the manuscript:

1.   The authors state that the use of immune checkpoint inhibitors (ICIs) as therapeutic agents might be a promising approach to the treatment of brain cancer. Thus, I think that in section 1 – Introduction, brain cancer and brain metastasis should be briefly addressed before introducing ICIs and their therapeutic relevance.

2.   In section 2.2. (line 97), which criteria were taken into consideration regarding animals' age and sex? Also, FUS sonication parameters (line 130) such as frequency were optimized?

3.   MRI images (Figures 2 and 4) present poor quality. Please verify the resolution of these figures.

4.   In section 4, discuss the potential applications, identify limitations, and future research directions.

5.   In section 4 (lines 223-226) please add the proper reference.

Author Response

Responses to Reviewer #4

Comment 1: The authors state that the use of immune checkpoint inhibitors (ICIs) as therapeutic agents might be a promising approach to the treatment of brain cancer. Thus, I think that in section 1 – Introduction, brain cancer and brain metastasis should be briefly addressed before introducing ICIs and their therapeutic relevance.

Response:

Following the reviewer's suggestion, we added a brief introduction of brain cancer and brain metastasis in the Introduction.

Manuscript Revision:

“It is estimated that annually, over 300,000 individuals are diagnosed with primary brain tumors, with an even larger number affected by brain metastases. Among the recent groundbreaking advancements, immune checkpoint inhibitor (ICI) therapy has emerged at the forefront, offering a promising avenue to bolster the body's own defense mechanisms against tumor cells.” [Lines 30-34]

Comment 2: In section 2.2. (line 97), which criteria were taken into consideration regarding animals' age and sex? Also, FUS sonication parameters (line 130) such as frequency were optimized?

Response:

The age of the pig was selected to be 4 weeks. No female pigs were used in this study. Future studies could look into the influence of age and sex but beyond the focus of this study. The FUS sonication parameters were selected in reference to our previous publications:

Xu, L.; Pacia, C. P.; Gong, Y.; Hu, Z.; Chien, C. Y.; Yang, L.; Gach, H. M.; Hao, Y.; Comron, H.; Huang, J.; Leuthardt, E. C.; Chen, H., Characterization of the Targeting Accuracy of a Neuronavigation-Guided Transcranial FUS System In Vitro, In Vivo, and In Silico. IEEE Trans Biomed Eng 2023, 70 (5), 1528-1538

Pacia CP, Zhu L, Yang Y, Yue Y, Nazeri A, Gach HM, Talcott MR, Leuthardt EC, Chen H*, Feasibility and safety of focused ultrasound-enabled liquid biopsy in the brain of a porcine model, Scientific Reports, 2020; 10: 7449

Manuscript revisions:

Added these two references to the FUS sonication parameters.

Comment 3: MRI images (Figures 2 and 4) present poor quality. Please verify the resolution of these figures.

Response:

These MRI images indeed have low resolution. We have provided the resolution of these images in the manuscript. [Lines 150-158]

Comment 4: In section 4, discuss the potential applications, identify limitations, and future research directions.

Response:

Following the reviewer’s suggestion, we have added discussions on these topics.

Manuscript Revision:

“This study demonstrated the feasibility and safety of FUS-BBBO in a large animal model, which supports the clinical translation of this technique for the delivery of ICIs. Although promising, the study has several limitations. First, this feasibility study used healthy pigs as the animal model. Future studies could further evaluate the drug delivery efficiency and efficacy using a pig model of brain tumor. Second, this study performed single-point sonication in each individual FUS sonication. Future studies need to implement large-volume sonication in order to cover the entire volume of the brain tumor.” [Lines 272-278]

Comment 5: In section 4 (lines 223-226) please add the proper reference.

Response:

Reference is added.

Manuscript Revision:

“Lee, H.; Guo, Y.; Ross, J. L.; Schoen, S., Jr.; Degertekin, F. L.; Arvanitis, C., Spatially targeted brain cancer immunotherapy with closed-loop controlled focused ultrasound and immune checkpoint blockade. Sci Adv 2022, 8 (46), eadd2288.”

Reference: 23

Round 2

Reviewer 1 Report

None